# FORGET-ME-NOT: LEARNING TO FORGET IN TEXT-TO-IMAGE DIFFUSION MODELS

## ABSTRACT

The unlearning problem of deep learning models, once primarily an academic concern, has become a prevalent issue in the industry. The significant advances in text-to-image generation techniques have prompted global discussions on privacy, copyright, and safety, as numerous unauthorized personal IDs, content, artistic creations, and potentially harmful materials have been learned by these models and later utilized to generate and distribute uncontrolled content. To address this challenge, we propose **Forget-Me-Not**, an efficient and low-cost solution designed to safely remove specified IDs, objects, or styles from a well-configured text-to-image model in as little as 30 seconds, without impairing its ability to generate other content. Alongside our method, we introduce the **Memorization Score (M-Score)** and **ConceptBench** to measure the models' capacity to generate general concepts, grouped into three primary categories: ID, object, and style. Using M-Score and ConceptBench, we demonstrate that Forget-Me-Not can effectively eliminate targeted concepts while maintaining the model's performance on other concepts. Furthermore, Forget-Me-Not offers two practical extensions: a) removal of potentially harmful or NSFW content, and b) enhancement of model accuracy, inclusion and diversity through **concept correction and disentanglement**. It can also be adapted as a lightweight model patch for Stable Diffusion, allowing for concept manipulation and convenient distribution. To encourage future research in this critical area and promote the development of safe and inclusive generative models, we will open-source our code and ConceptBench.

## 1 INTRODUCTION

In recent advancements, text-to-image models (Chang et al., 2023; Gafni et al., 2022; Ramesh et al., 2021; 2022; Saharia et al., 2022; Yu et al., 2022; Rombach et al., 2022; Xu et al., 2022) have exhibited remarkable capabilities in generating high-resolution images based on textual descriptions. Notably, diffusion models, exemplified by DALL-E 2 (Ramesh et al., 2022) and Stable Diffusion (Rombach et al., 2022), have satisfied commercial-grade productization standards, paving the way for a plethora of applications tailored for end-users. Furthermore, recent literature has evidenced the potential of these models to produce and modify videos in a zero-shot manner(Khachatryan et al., 2023), eliminating the need for additional training. Industrial implementations, as referenced in Picsart; Firefly; Lexica; NovelAI; Midjourny; Somepalli et al. (2022), have garnered widespread acclaim in the domains of art and visual design, thereby attracting significant public interest.

However, the burgeoning popularity of this domain has concurrently raised pertinent concerns encompassing security, fairness, regulatory compliance, intellectual property rights, and overall safety. The community faces pressing challenges, such as the inadvertent generation of unauthorized, prejudiced, and potentially hazardous content. This is not an unprecedented issue, as the academic community has previously endeavored to address similar concerns.

Figure 1: **Concept Forgetting:** target concepts (denoted in blue text and crossed-out) are successfully removed without compromising the quality of the output. **Concept Correction & Disentangle:** our method can be used to correct a dominant or undesired concept of a prompt. Prior overshadowed concepts reveal in outputs after the dominant concepts are forgotten.

The inherent risks associated with large-scale text-to-image models predominantly stem from the vast datasets employed during their training phase. These datasets, which include public repositories like Laion (Schuhmann et al., 2022), COYO (Byeon et al., 2022), CC12M (Changpinyo et al., 2021), and proprietary data from renowned entities such as Google (Saharia et al., 2022; Yu et al., 2022) and OpenAI (Ramesh et al., 2021; 2022), present unique challenges. Public datasets, often sourced from web scrapes, may lack rigorous quality control measures, especially concerning bias and safety. Conversely, proprietary datasets, while potentially more controlled, are constrained by scalability issues due to the inherent costs of annotation. Consequently, mitigating issues related to harmful content, privacy breaches, and copyright infringements through mere data filtration or source attribution becomes a daunting task. One potential recourse could be domain adaptation (Giannone et al., 2022; Zhu et al., 2022; Xiao et al., 2022). However, the intricacies of curating and refining such datasets remain formidable. Moreover, such domain adaptation can inadvertently diminish the model's versatility, rendering it inept at synthesizing out-of-domain images.

Therefore, efficient methods that enable large-scale text-to-image models to selectively *omit specific concepts* emerge as a promising direction. A line of concurrent works explore this direction by overwriting model predictions of a target concept to be something else (Gandikota et al., 2023; Kumari et al., 2023). Although they are effective in erasing inappropriate generations relating to target concept, they are essentially training the model to learn a remapping between target concept and manually defined generations, rather than letting the model fall back to whatever it generates in light of its own knowledge, after omitting a concept. We argue that it is desiderative to omit target concept by delving in the conditioning mechanism which is the ultimate trigger of a concept appearing in generations. Moreover, cross attention conditioning exhibits promising results in recent controllable text-to-image synthesis works (Chefer et al., 2023; Hertz et al., 2022).

This paper embarks on its journey by delineating this novel paradigm, termed *concept forgetting*. In pursuit of this objective, we observe that scores of image-to-text cross attention conditioning can be directly used as objectives to optimize diffusion model for weakening the model perception of target concepts. To the best of our knowledge, we are the first to show cross attention scores are viable objectives for fine-tuning text-to-image models. On top of this technique, we introduce **Forget-Me-Not**, a cost-effective methodology for concept omission where a model decides what to

generate afterwards based on its knowledge. Complementing this, we present the **memorization score (M-score)** in tandem with **ConceptBench**. It quantifies a model's prowess in generating specific concepts while the latter establishes benchmarks for evaluating concept retention and omission. Furthermore, we augment the realm of concept forgetting with *concept correction & disentanglement*, enhancing model precision and diversity.

In summation, this paper's primary contributions are encapsulated as follows:

- The introduction of **Forget-Me-Not**, a cross-attention-based, efficient methodology for concept omission and rectification in large text-to-image models. This approach not only facilitates rapid concept omission in a mere 30 seconds but also seamlessly integrates as lightweight patches for Stable Diffusion, ensuring multi-concept manipulation and user-friendly distribution.

- The inception of the **memorization score (M-score)** and **ConceptBench**, pioneering quantitative tools that assess models' proficiency in concept synthesis and omission.

- Comprehensive empirical evaluations underscore the simplicity, cost-effectiveness, and efficacy of our method. Its applicability extends to the removal of objectionable content and the rectification of biased concepts, aligning more closely with real-world applications.

## 2 RELATED WORKS

### 2.1 TEXT-TO-IMAGE SYNTHESIS

In the past decade, we have witnessed the rapid advance of it from unconditional generative models to conditional generative models with powerful architectures of auto-regressive model (Ramesh et al., 2021; Yu et al., 2022), GAN (Casanova et al., 2021; Karras et al., 2019; 2021; Walton et al., 2022; Shaham et al., 2019) and diffusion process (Ho et al., 2020; Nichol & Dhariwal, 2021; Lu et al., 2022; Dockhorn et al., 2022; Balaji et al., 2022; Song et al., 2020). Early works focus on unconditional, single-category data distribution modeling , such as hand-written digits, certain species of animals, and human faces (Deng, 2012; Choi et al., 2020; Karras et al., 2019; Liu et al., 2018). Though, unconditional models quickly achieves photo realistic results among single-category data, it's shown that mode collapsing issue usually happens when extending data distributions to multiple-category or real image diversity (Casanova et al., 2021; Metz et al., 2016; Arjovsky et al., 2017). To tackle the model collapsing problem, the conditional generative model has been introduced. Various types of data have been used as the conditioning for generative models, e.g. class labels, image instances, and even networks (Casanova et al., 2021; Mirza & Osindero, 2014) etc. At the same time, CLIP (Radford et al., 2021; Ilharco et al., 2021), a large-scale pretrained image-text contrastive model, provides a text-image prior of extremely high diversity, which is discovered to be applicable as the conditioning for generative model (Nichol et al., 2021; Crowson et al., 2022; Liu et al., 2023). Nowadays, DALL-E 2 (Ramesh et al., 2022) and Stable Diffusion (Rombach et al., 2022) are capable of generating high quality images solely conditioning on free-form texts, Subsequently, a line of work seeks to efficiently adapt the massive generative model to generate novel rendition of an unseen concept represented by a small reference set. Dreambooth (Ruiz et al., 2022) adapts the model by finetuning all of its weights, while it requires enormous storage to save newly adapted weights. Textual Inversion (Gal et al., 2022) and LoRA (Hu et al., 2021) ameliorate the issue by adapting the model by adding a small set of extra weights.

### 2.2 CONCEPT OVERWRITING

Prior works have noticed the inadvertent biased and unsafe generations from large text-to-images models. They adopt the denoising loss (Dhariwal & Nichol, 2021) to steering predicted noise away from what is used to be for a given concept. Kumari et al. (2023) and Heng & Soh (2023) choose to overwrite target concept to manually defined anchor concept, for example overwrite the prediction of "a grumpy cat" to "a cat". SLD (Schramowski et al., 2023) and ESD (Gandikota et al., 2023) utilize the classifier-free guidance of a pretrained model to steer prediction of a trainable model to the opposite guidance direction. As a result, target concept is overwritten by that opposite guidance.

## 3 METHODS

### 3.1 PRELIMINARIES

**Diffusion models** (Ho et al., 2020; Nichol & Dhariwal, 2021; Dhariwal & Nichol, 2021) are denoising models that iteratively restore data $x_0$ from its Gaussian noise corruption $x_T$ with a total step number $T$. Such a restoration process is usually known as the reverse diffusion process $p_\theta(x_{t-1}|x_t)$ and the opposite of the reverse process is the forward diffusion process that blends the signal with noise $q(x_t|x_{t-1})$:

$$q(x_t|x_{t-1}) = \mathcal{N}(x_t; \sqrt{1-\beta_t}x_{t-1}; \beta_t \mathbf{I})$$
$$p_\theta(x_{t-1}|x_t) = \mathcal{N}(x_{t-1}; \mu_\theta(x_t, t); \Sigma_\theta(x_t, t))$$

Both forward and reverse processes are presumably Markovian chains, so we can express the likelihood of both processes as:

$$q(x_{1:T}|x_0) = \prod_{t=1}^{T} q(x_t|x_{t-1}) \qquad p_\theta(x_{0:T}) = p(x_T)\prod_{t=1}^{T} p_\theta(x_{t-1}|x_t)$$

The loss function for the diffusion process is then to minimize the variational bound $\mathcal{L}_{vlb}$ of the negative log-likelihood $p_\theta(x_0)$ (maximize the likelihood of $x_0$ as the final denoised result from a model with parameters $\theta$):

$$\mathcal{L}_{\text{VLB}} = \mathbb{E}\left[-\log p_\theta(x_0)\right] \leq \mathbb{E}_q\left[-\log \frac{p_\theta(\mathbf{x}_{0:T})}{q(\mathbf{x}_{1:T}|\mathbf{x}_0)}\right]$$

**Cross-Attentions** (Vaswani et al., 2017) are widely adopted deep learning modules used in discriminative models (Dosovitskiy et al., 2020; Carion et al., 2020; Jain et al., 2022), conditional generation models (Ramesh et al., 2022; Saharia et al., 2022; Rombach et al., 2022) as well as language models (Devlin et al., 2018; Raffel et al., 2020; Chung et al., 2022). The purpose of cross-attention is to transfer information from conditional inputs to hidden features through dot product and softmax. For example, in stable diffusion (Rombach et al., 2022), the hidden feature serves as the query $Q$ and context serves as key $K$ and value $V$. Assume $Q$ and $K$ has dimension $d$ for inner product, the output $h$ is then computed as the following:

$$h = \text{softmax}(\frac{QK^T}{\sqrt{d}})V$$

### 3.2 CONCEPT FORGETTING

A concept is an abstract term representing an intuited object of thought, which also serves as the foundation for people's perceptions. Specifically for generative task, we may recognize concepts as tangible things, including identities, objects that physically existed, style of images, object relations, and even poses and behavior. On contrast to machine unlearning (Baik et al., 2020), which aims to delete the fields around designated data points, we define concept forgetting in diffusion models as the disentanglement of concept prompts and visual contents. This definition allows models to retain their generative abilities to the greatest extent possible.

### 3.3 FORGET-ME-NOT

To fulfill the aforementioned goals in Section 3.2, we introduce **Forget-Me-Not**. Forget-Me-Not is well-capable in removing a wide variety of concepts without inadvertently influencing too much on other outputs. Its underlying methodology, *attention resteering*, fits almost all major text-to-image models and may extend to other conditional multimodal generative models.

**Cross Attention Conditioning** As discussed before, the text conditioning happens at each cross attention layer of UNet, where visuals of a concept is first introduced to hidden features through QKV attention operation. A logical try is to manually set the attention from hidden features towards tokens of target concept as zero. This simple manipulation turns out successful that expected visuals

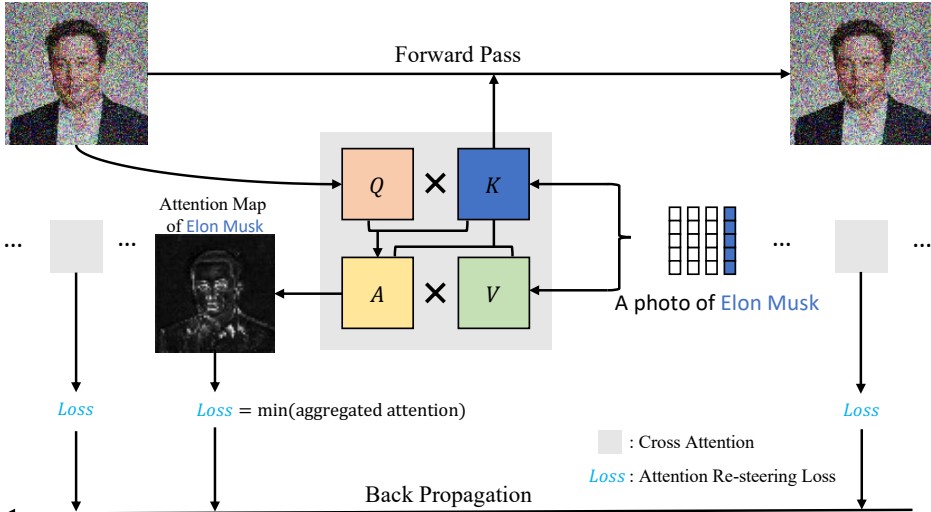

Figure 2: This figure shows the Attention Re-steering we proposed in our Forget-Me-Not method, in which we set the objective function to minimize the attention maps of target concepts (Elon Musk in this case) and correspondingly finetune the network.

of target concept are diminishing in result images. Following this observation, we devise Attention Resteering Loss to backpropagate the concept diminishing effect to model weights.

**Attention Resteering Loss** conceptually enables a precise backpropagation from exact tokens of target concept. Figure 2 shows the diagram of this idea, in which we first locate the token embeddings associated with target concept; compute the attention maps between visual hidden features and token embeddings; aggregate corresponding attention scores; minimize the aggregated score and backpropagate the network. Such attention resteering can be plugged into any cross-attention layers of the network. It also decouples model's finetuning from its original loss functions(variational lower bound for diffusion models), making the process a much simpler solution. One of our main focuses in this work is concept forgetting of text-to-image models; therefore we carry out attention resteering on all cross-attention layers of UNet (Ronneberger et al., 2015) in Stable-Diffusion (SD) (Rombach et al., 2022), which yields the best performance on most of the concepts. The entire algorithm of Forget-Me-Not can be found in Algorithm 1.

---

**Algorithm 1** Forget-Me-Not on diffuser

---

**Require:** Token embeddings $\mathcal{C}$, embedding locations $\mathcal{N}$, and reference images $\mathcal{R}$ of target concept, diffuser $G_\theta$, diffusion step $T$, optimization steps $\mathcal{S}$.

1: **repeat** $\mathcal{S}$ steps
2: $\quad t \sim \text{Uniform}([1 \dots T]); \epsilon \sim \mathcal{N}(\mathbf{0}, \mathbf{I})$
3: $\quad r_i \sim \mathcal{R}; c_j, n_j \sim \mathcal{C}, \mathcal{N}$
4: $\quad x_0 \leftarrow r_i$
5: $\quad x_t \leftarrow \sqrt{\bar{\alpha}_t} x_0 + \sqrt{1 - \bar{\alpha}_t} \epsilon$
6: $\qquad\qquad\qquad\qquad\qquad\qquad\qquad\qquad\qquad \triangleright \bar{\alpha}_t$: noise variance schedule
7: $\quad x'_{t-1}, A_t \leftarrow G_\theta(x_t, c_j, t)$
8: $\qquad\qquad\qquad\qquad\qquad\qquad\qquad\qquad\qquad \triangleright A_t$: all cross attention maps
9: $\quad \mathcal{L} \leftarrow \sum_{a_t \in A_t} \|a_t^{[nj]}\|^2$
10: $\qquad\qquad\qquad\qquad\qquad\qquad\qquad\qquad\qquad \triangleright \mathcal{L}$: attention resteering loss
11: $\quad \theta \leftarrow \theta - \nabla_\theta \mathcal{L}$

---

**Optional Concept Inversion:** Although we may directly obtain token embeddings using prompts for most text-to-image models, this is not a generalized case for all concepts, particularly when it's hard or impossible to describe a concept using textual prompts. For instance, forget a style represented by an image. To overcome this challenge, we optionally include the textual inversion (Gal et al., 2022) as a fixed overhead before Forget-Me-Not to strengthen its generality on all concepts.

In practice, we also notice that such inversion helps text-to-image models more precisely identify the forgetting concept and thus improves their performance. More results can be found in our Experiments Section. Ablation studies on the effects of Concept Inversion(CI) are in Appendix C.

# 4 EXPERIMENTS

## 4.1 EXPERIMENTAL SETUP

**Dataset:** Our evaluation utilizes datasets constructed from ConceptBench. Each dataset comprises pairs of images and their corresponding textual prompts and they are grouped by concept category, with images generated via the Stable Diffusion 2.1 base. See Appendix A. Implementation details are in Appendix B.

**Baselines:** Our approach is benchmarked against other concurrent works, ESD (Gandikota et al., 2023) and ACTD (Kumari et al., 2023). The comparison focuses on forgetting performance and the influence on related concepts, as detailed in section 4.2. Our unique capability for concept correction is highlighted in section 4.4.

**Evaluation Metrics:** Forgetting efficacy is assessed using both the CLIP score (Radford et al., 2021) and M-Scores. The CLIP score measures the congruence between generated images and their textual prompts. A decrease in the CLIP score signifies effective forgetting, but an excessively low score indicates a loss of overall image semantics, see Table 1. For instance, after forgetting the "corgi" concept, we anticipate the generated image to still represent a dog, not an unrelated object like a metal bucket.

The M-Score offers a model-centric perspective on forgetting. It employs an anchor concept in the CLIP space to gauge the similarity between the inverted concept and the anchor across both the original and forgetting models. Further insights on this are provided in the subsequent Memorization Measurement section.

**Ablation Studies:** We delve into the impact of various configurations, including different sets of trainable weights and the utilization of inverted concepts from images. Comprehensive details are available in the Appendix C.

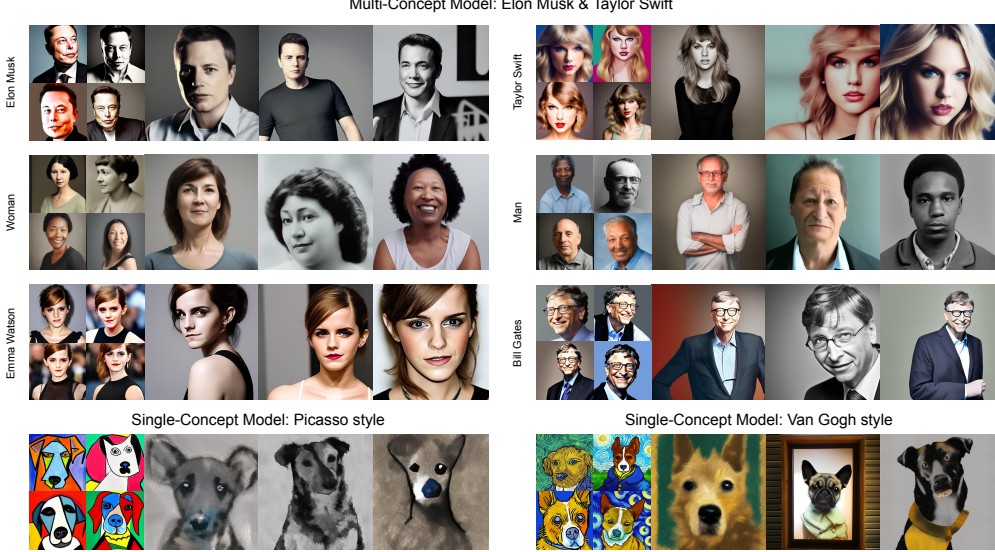

Figure 3: Concept forgetting results using our method. The initial 2x2 grid displays original samples from Stable Diffusion. Following this, three images depict post-forgetting samples generated from the same prompt. The top three rows, targeting Elon Musk and Taylor Swift, highlight our multi-concept forgetting capability. Control concepts, including Bill Gates and Emma Watson, illustrate the limited influence our method has on non-targeted concepts. The final row presents single-concept style models. Image prompts used were: "a photo of X" (for the first three rows) and "a dog in X style" (for the last row).

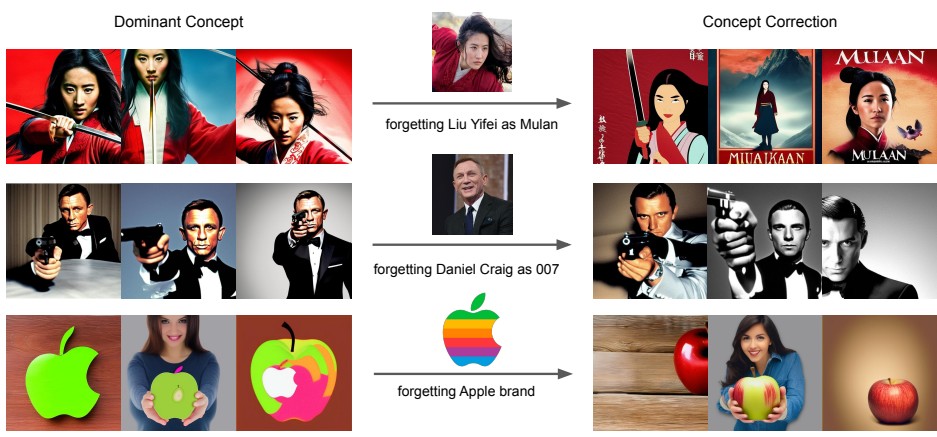

Figure 4: Concept Correction: Our method's effectiveness in diminishing dominant concepts allows for the emergence of secondary concepts within semantically-rich prompts. The displayed images, generated from top to bottom, correspond to the prompts: "a movie poster of Mulan", "James Bond", and "apple shape".

## 4.2 QUALITATIVE COMPARISON

Figure 3 showcases the partial results of our concept forgetting benchmark. The multi-concept model, targeting Elon Musk and Taylor Swift, exemplifies our method's proficiency in multi-concept forgetting. Notably, the first row reveals the successful forgetting of both target concepts. Our evaluation also considered the repercussions of forgetting on related concepts, specifically man, woman, Bill Gates, and Emma Watson. The results indicate that our Forget-Me-Not method preserves content and maintains visual quality effectively. However, subtle changes in pose and style were observed for the man and Bill Gates concepts. These observations suggest that our method might influence closely associated concepts more than distant ones. The final row further highlights the emergence of a new painting style post-forgetting the styles of Picasso and Van Gogh.

In comparison to alternative methods (Gandikota et al., 2023; Kumari et al., 2023), our approach exhibits superior generative capabilities for related concepts. ESD (Gandikota et al., 2023), which employs inverse concept-free guidance, adversely affects related concepts. Conversely, ACTD (Kumari et al., 2023) adjusts the forgetting concept relative to an anchor concept, making its performance contingent on the choice of anchor. This becomes challenging for abstract concepts where an apt anchor is elusive.

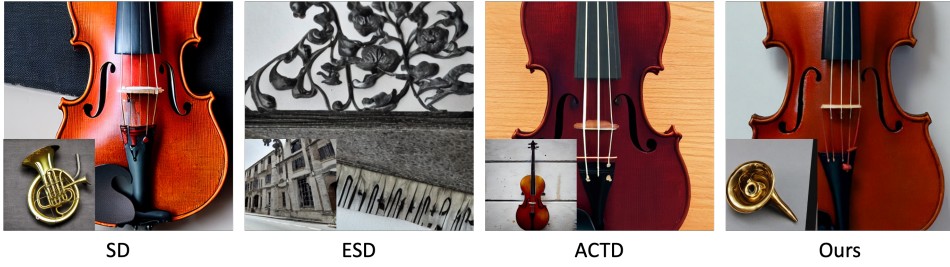

| SD | ESD | ACTD | Ours |

Figure 5: Comparison of methods for forgetting the french horn concept. Each primary figure is prompted with "a photo of a violin", accompanied by a smaller "a photo of a french horn" image. To fairly compare with others, we show results on SD 1.5 in this figure. ESD not only disrupts the french horn concept but also adversely affects the related violin concept. ACTD, using "instrument" as the anchor concept, which is dominated by the violin in the SD model, results in the french horn concept merging with the violin concept. In contrast, our method selectively removes the french horn features while preserving the salient features of other related concepts.

A photo of Mango

Stable Diffusion      Negative Prompt: dog      Ours

Figure 6: In concept correction, our method has the advantage of comprehensive forgetting over negative prompt. In this example, we also tests "animal" as negative prompts, yet it still generates dogs/cats.

## 4.3 QUANTITATIVE ANALYSIS

**Memorization Measurement** We employ textual inversion (Gal et al., 2022) to pinpoint token embeddings that align closely with images. This technique aids in assessing the shifts in inverted embeddings of anchor images relative to a reference, both pre and post-forgetting. These shifts reflect the generative model's retention level of a concept, termed as Memorization Score.

For instance, with the prompt "Elon Musk", its concept embedding ($\mathbf{emb}_r$) is derived from a text encoder. Anchor images of Elon Musk are then inverted using both the *original model* and the *forgetting model*. The resulting embeddings, original textual inversion ($\mathbf{emb}_o$) and forgetting textual inversion ($\mathbf{emb}_f$), are compared to gauge the change in the embedding space. Only the **pooler tokens** of concept embeddings are used for measurement. The change in concept embedding is quantified as the difference between $\cos(\mathbf{emb}_r, \mathbf{emb}_o)$ and $\cos(\mathbf{emb}_r, \mathbf{emb}_f)$. A decline signifies effective forgetting. Due to the inherent variability in the textual inversion process, we compute the Memorization Score's average over five iterations. Table 1 presents the scores for each sub-category.

Table 1: Memorization Scores and CLIP Scores of instances from each sub-categories.

| Category | Concept | Initial M-Score | Forgetting M-Score ↓ | Initial CLIP Score | Forgetting CLIP Score ↓ |
|---|---|---|---|---|---|
| Person | Elon Musk | 0.943 | 0.848 | 0.308 | 0.285 |
| Animation | Mickey Mouse | 0.948 | 0.836 | 0.304 | 0.269 |
| Animal | Zebra | 0.972 | 0.899 | 0.312 | 0.310 |
| Object | Google | 0.940 | 0.811 | 0.216 | 0.209 |
| Fruit | Apple | 0.696 | 0.493 | 0.267 | 0.258 |
| Animal | Horse | 0.877 | 0.808 | 0.275 | 0.266 |
| Style | Van Gogh | 0.916 | 0.684 | 0.274 | 0.233 |

## 4.4 CONCEPT CORRECTION

Text-to-image models often prioritize the semantics of a prompt based on the abundance of image-text examples during training. This can overshadow less prevalent semantics during inference, as illustrated in Figure 4. For instance, the James Bond series predominantly showcases Daniel Craig. Our method, however, can reduce the dominance of such a semantic, enabling visibility of other James Bond actors. Similarly, our approach effectively rectifies target concepts in scenarios where semantics compete, as seen with the Mulan series and the term "apple".

Negative prompts, used in text-to-image synthesis to exclude unwanted concepts, can inadvertently alter other image attributes. Moreover, they may not always rectify undesired concepts. As depicted in Figure 6, the prompt "a photo of a mango" often yields dog images due to the popularity of "mango" as a pet name. Our method adeptly retrieves the mango fruit by dissociating the prompt from dog/cat images.

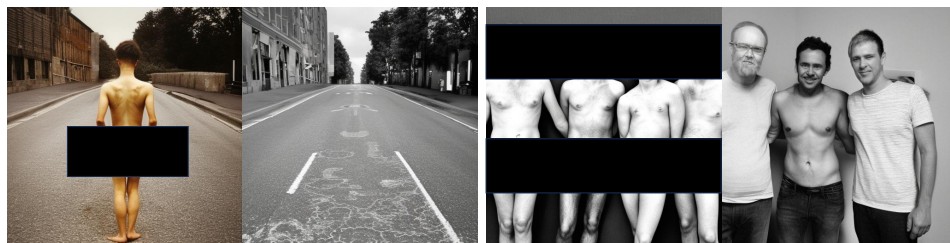

Figure 7: Results of removing NSFW contents triggered by "naked". Faces and sensitive parts are blacked out.

## 4.5 NSFW Removal

We evaluate our method's capability to eliminate inappropriate content, denoted as NSFW ("not safe for work"). Such content, potentially offensive even to adults, can inadvertently be part of large datasets like LAION (Schuhmann et al., 2022), despite the use of NSFW detectors (LAION-AI). Stable Diffusion, trained on LAION, has been known to produce NSFW content with specific prompts.

To assess our approach, we utilize a known NSFW-triggering prompt, "a photo of naked", in the Stable Diffusion v2.1 base model. Using this setup, the model consistently produces inappropriate images. We then train Forget-Me-Not using eight of these NSFW images.

The outcomes, presented in Figure 7, confirm the successful forgetting of the "naked" concept. The original NSFW images have undergone black modifications to ensure appropriateness. Our method effectively forgets NSFW content without requiring extra data or third-party NSFW detectors.

## 5 Conclusion

In this study, we investigate concept forgetting in text-to-image generative models and introduce Forget-Me-Not. This lightweight approach enables ad-hoc concept forgetting using only a few either real or generated concept images; it can also be easily distributed using model patches. Forget-Me-Not is further naturally extended to enable concept correction and disentanglement. Our experiments demonstrate that Forget-Me-Not is successful in diminishing and correcting target concepts in Stable Diffusion. Additionally, we introduce ConceptBench and Memorization Score as evaluation metrics. Overall, our work provides a foundation for further research on concept forgetting and manipulation in text-to-image generation, and can be further extended to other conditional multimodal generative models to improve the accuracy, inclusion and diversity of such models.

## 6 Social Impact & Limitations

**Social Impact** Our research has a positive social impact by offering an effective and cost-efficient method to remove and correct harmful and biased concepts in text-to-image generative models. These models are rapidly becoming the backbone of popular AI art and graphic design tools, used by a growing number of people. Our method can generate lightweight model patches that can be conveniently distributed to text-to-image model users like how conventional software patch works. Thus, our research takes a small step towards promoting fairness and privacy protection in AI tools, ultimately benefiting society as a whole.

**Limitations** While our approach performs well on concrete concepts in ConceptBench, it faces challenges in identifying and forgetting abstract concepts. Additionally, successful forgetting may require manual interventions, such as concept-specific hyperparameter tuning.

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

## A   DATASET

To meet our need to evaluate Forget-Me-Not and potential future concept forgetting approaches, we introduce a benchmark, namely ConceptBench. It is important to note that several existing benchmarks, such as Lin et al. (2014); Saharia et al. (2022), help assess overall generation quality. However, none are specifically designed to measure a model's ability to memorize and forget. ConceptBench utilizes instances from LAION (Schuhmann et al., 2022), forming three categories,identity, object, and style, ranging from discrete to abstract and easy to hard.

Identity refers to the unique and discrete features of each instance. Specifically, we examine identity in subcategories such as person, franchise, animal, and brand. The person and franchise subcategories are of particular interest due to potential privacy or copyright issues associated with generated images of celebrities or intellectual properties. For the animal breed subcategory, detailed instances may include specific breeds, such as "Corgi", which belong to the more general "dog" category but have distinct visual features. In the case of brands, they represent abstract concepts of intangible objects that can manifest as logos throughout our daily lives.

Object is a broader concept encompassing multiple variations. By combining identity instances mentioned earlier, this category provides a hierarchical structure to examine the influence of concept forgetting on the model. We include food items like "apple", "banana", and "broccoli", man-made objects such as "airplane", and general animals like "dog" and "cat."

Style is an abstract concept that determines the overall appearance of generated images. ConceptBench incorporates styles such as "Van Gogh", "Claude Monet". See Figure 11 for extra results.

## B   IMPLEMENTATION DETAILS

We run our experiments on Stable Diffusion 2.1 base with its default generations settings at inference time: 50 steps, guidance strength 7.5, EulerDiscreteScheduler. For training, we use batch size 1, learning rate 2e-6 and optimization steps 35 as our base setting. On single A100 GPU, this optimization process takes around 30 seconds.

## C   ABLATION STUDIES

**Inverted Concept v.s. Textual Concept** We conducted controlled experiments on forgetting with and without using concept inversion (CI), which is detailed at the end of Section 3.3. Concept inversion is used to handle concepts that are difficult to describe using prompts. Generally, it can help extract the target concept from a set of images, resulting in arguably more precise embeddings. We first choose two concepts that can be well described using textual prompt: Elon Musk and a man. Then, we generate a small set of images to get their inverted concepts. Our results show that inverted concepts can achieve higher fidelity as illustrated in Figure 8, where the model trained with CI preserved more of the original poses and details.

Here we demonstrate a failed case of CI. The downside of using CI is hard to ensure the quality of inverted concepts, which may not embed the target concept as expected. The quality of inverted concept depends on a variety of factors such as image and prompt used for inversion, and additional set of hyper-parameters for CI. In Figure 9, we demonstrate a situation where textual concept prevails inverted concept. By using the same settings except for token embeddings, textual concept "airplane" succeeds while inverted tokens fails. We hypothesize the some unexpected information leaks into inverted concepts.

**Trainable Weights** We compare finetuning the entire UNet model versus only the cross attention (CA) layers. Cross attention is a critical component for text-to-image generation, as it injects textual information into the image formation process. Given the same hyper-parameter settings except for steps, our results show that both methods can successfully achieve concept forgetting. However, finetuning the entire UNet model tended to break the model's generation capability in fewer steps. In some cases, the model collapsed before the forgetting process was complete, as show in the "Broccoli" case of Figure 10. Note that we use Elon Musk as a control concept at the last column, demonstrating the influence of forgetting of target concept to unrelated concepts. Results show that Elon Musk has been kept intact after forgetting by just tuning cross attention layers.

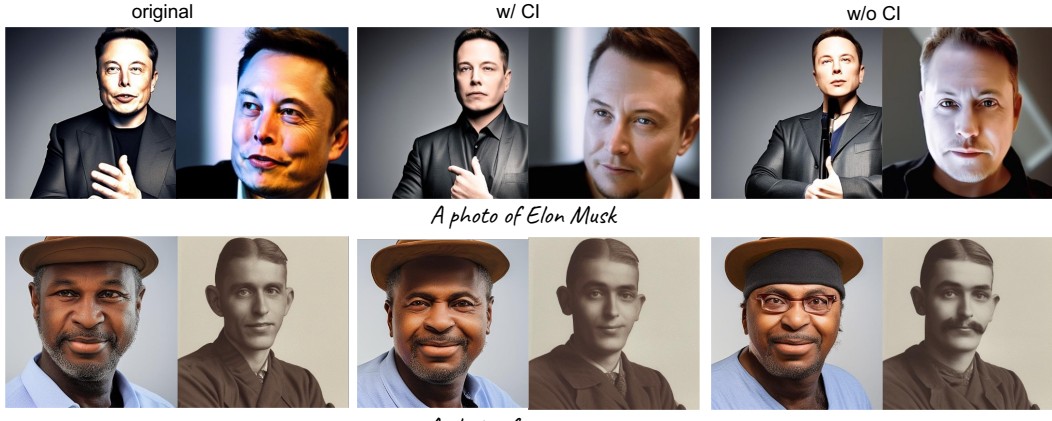

Figure 8: An example that inverted tokens extract precise semantics into dedicated tokens, allowing for better pose and feature consistency.

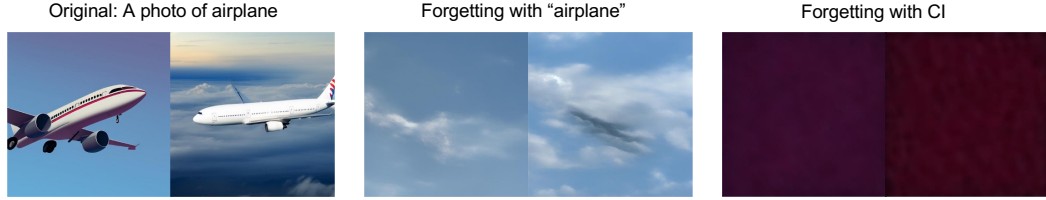

Figure 9: An example of inverted tokens failing to capture target concept precisely.

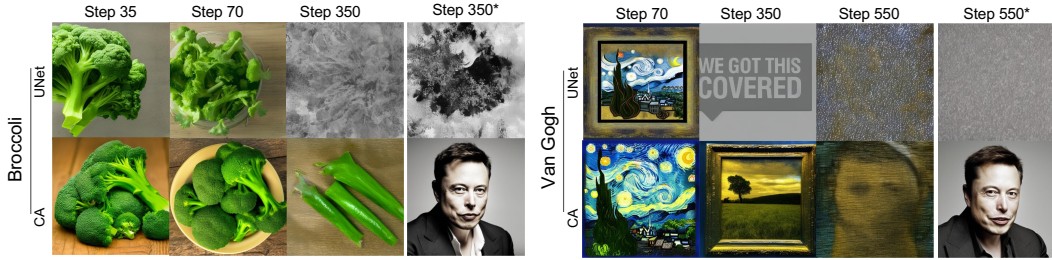

Figure 10: Trainable weights ablation comparing UNet and Cross Attention (CA). The last column with Step X* shows the control concept Elon Musk at Step X.

Stable Diffusion | Forget-Me-Not

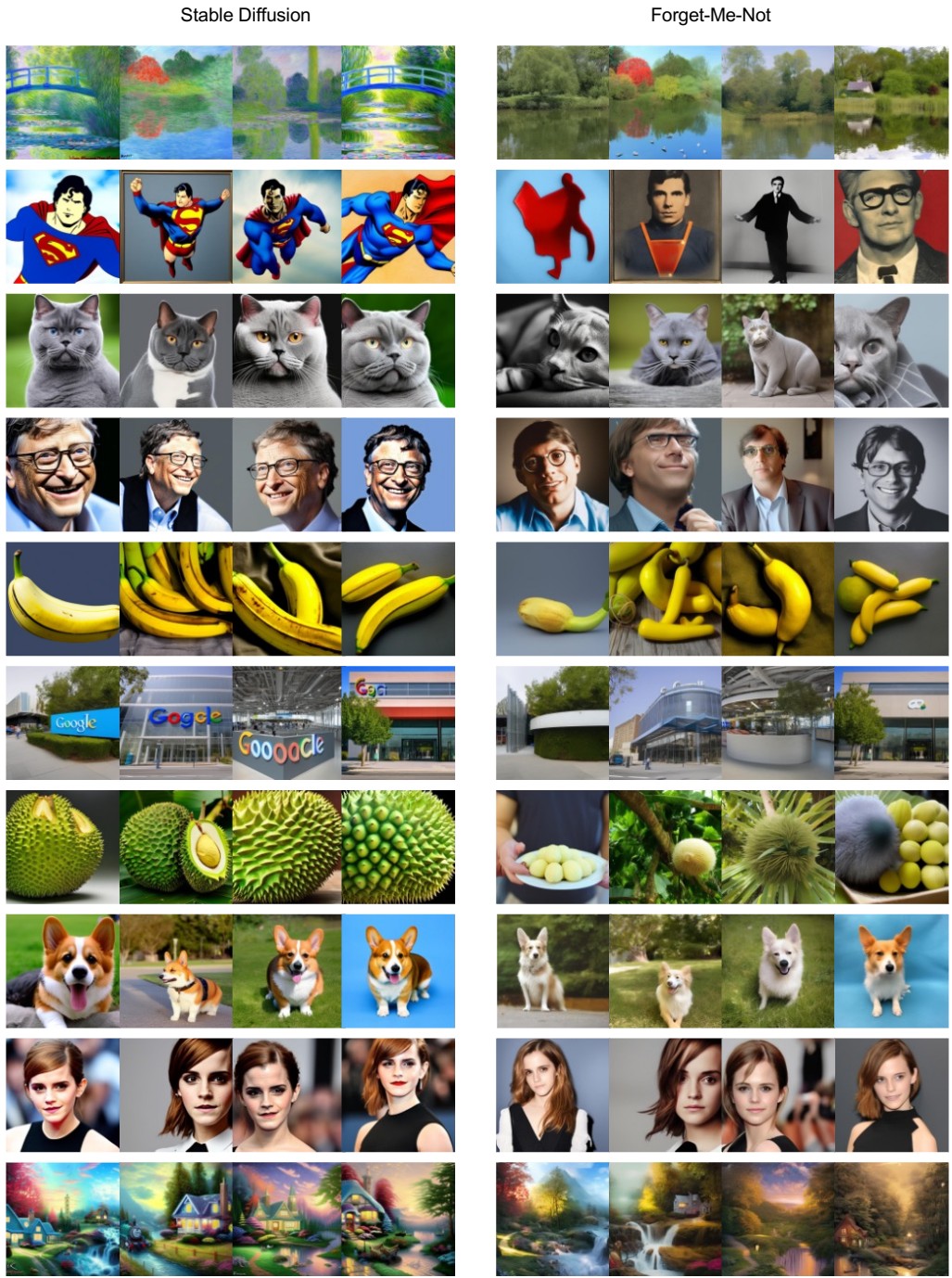

Figure 11: Extra results from our method. From top to bottom: Claude Monet Style, superman, British short-hair, Bill Gates, banana, Google, durian, corgi, Emma Watson, Thomas Kinkade style

