# OpenReview forum: "Forget-Me-Not: Learning to Forget in Text-to-Image Diffusion Models"
_ICLR.cc/2024/Conference — ICLR 2024 Conference Withdrawn Submission_

### Official Review · Reviewer_r3ea · 2023-10-27

**Soundness:** 1 poor
**Presentation:** 1 poor
**Contribution:** 3 good
**Rating:** 1
**Confidence:** 2

**Summary:**

n/a - The paper does not meet minimum standards for presentation quality.

**Strengths:**

n/a - The paper does not meet minimum standards for presentation quality.

**Weaknesses:**

The paper requires significant improvements in presentation and it is currently nearly impossible to understand its contribution.

The following need to be defined formally in the main body of the paper: the M-score, when a concept is considered forgotten (how are CLIP embeddings used precisely?), ConceptBench, the loss function used for erasure.

In addition, the paper reads more like a sequence of jotted down notes without logical connections between most paragraphs.

Overall, I believe the paper does not meet a minimum bar of presentability and, therefore, I cannot adequately review it and recommend a rejection.

**Questions:**

n/a - The paper does not meet minimum standards for presentation quality.

---

### Official Review · Reviewer_Tnea · 2023-10-29

**Soundness:** 2 fair
**Presentation:** 3 good
**Contribution:** 2 fair
**Rating:** 5
**Confidence:** 4

**Summary:**

In response to the challenge of forgetting specific concepts, the proposed Forget-Me-Not, a cost-effective and efficient solution. This tool is designed to safely eliminate designated IDs, objects, or styles from a well-tailored text-to-image model in just 30 seconds, all while preserving its capacity to generate other content. In conjunction with the approach, two new metrics: the Memorization Score (M-Score) and ConceptBench were introduced. These metrics evaluate the model's ability to generate broad concepts, categorized into three main groups: ID, object, and style. By using M-Score and ConceptBench, Forget-Me-Not effectively removes targeted concepts while keeping the model's performance intact for other concepts.

**Strengths:**

1. Easy to implement and low computational cost.
2. The introduction of the Memorization Score (M-score) and ConceptBench, quantitative instruments designed to evaluate the models' competency in both concept generation and exclusion.

**Weaknesses:**

1. There is no evaluation of CLIP scores and M-Scores of Previous methods such as ESD.
2. As there is no theoretical justification for the method proposed, only qualitative evaluation with previous benchmarks is not enough. More quantitative evaluations need to be reported.
3. More experimentation of different styles, and object removal should be done as shown in the previous methods.

**Questions:**

1. As per my understanding, the CLIP scores can be evaluated for the proposed method and ESD. Is there  any specific reason that it is not done?

---

### Official Review · Reviewer_hT3k · 2023-11-01

**Soundness:** 2 fair
**Presentation:** 2 fair
**Contribution:** 3 good
**Rating:** 5
**Confidence:** 5

**Summary:**

This paper introduces an innovative, cost-effective strategy for eliminating specific concepts or objects from text-to-image models. To assess the efficacy of the unlearned text-to-image models, the authors present the memorization score (M-score) and introduce the ConceptBench.

**Strengths:**

1. Exploring Concept Correction and Disentanglement offers a captivating avenue for advancing diffusion model unlearning.
2. The introduced method is both efficient and cost-effective, streamlining the process.
3. This paper presents a novel metric, the M-score, and unveil a new dataset tailored for evaluation purposes.

**Weaknesses:**

1. The manuscript lacks comprehensive descriptions of the algorithms, making it challenging to understand and replicate the proposed method.
2. While the paper asserts that the introduced method can remove the target concept without affecting other concepts, it does not provide any experimental evidence to support this claim.
3. The paper does not offer a quantitative analysis specifically on the nudity concept. I recommend utilizing NudeNet as a nudity detection tool and employing a subset of the I2P dataset for a more robust evaluation.

**Questions:**

There exist adversarial attacks targeting unlearned diffusion models. These attacks challenge the robustness of these models by asserting that they remain vulnerable. Specifically, attackers can manipulate the unlearned diffusion models to produce images with concepts purportedly unlearned. This is achieved by appending or inserting adversarial text tokens, thereby raising concerns about the model's safety and efficacy.

---

### Official Review · Reviewer_Kf75 · 2023-11-09

**Soundness:** 2 fair
**Presentation:** 2 fair
**Contribution:** 3 good
**Rating:** 5
**Confidence:** 3

**Summary:**

This work focuses on unlearning specific targeted concepts from large pre-trained diffusion models. The approach used by the paper minimizes the attention map activation for a target concept, and updates the parameters accordingly. Two metrics, M-score and CLIP score are used to measure how well the model has forgotten a concept. In addition, the work proposes a benchmark named ConceptBench to check how well models can forget specific concepts such as objects, identities and styles.

**Strengths:**

1. This work focuses on a timely and important topic. With ongoing debate regarding copyrights and regulation for generative models, how specific (copyrighted) concepts can be deleted from generative models is highly relevant.
2. The proposed approach seems reasonable and is easy to understand intuitively. The approach also seems to be faster, and less cumbersome than prior approaches that require other concepts as "anchors" [1].

[1] Kumari, Nupur, et al. "Ablating concepts in text-to-image diffusion models." _Proceedings of the IEEE/CVF International Conference on Computer Vision_. 2023.

**Weaknesses:**

1. The writing quality of the paper is not good.  Generally, the paper also uses difficult prose in several places and this decreases the quality of the paper significantly. Also, it would be good to provide references to the different sections using hyperlinks. There are several typos, confusing language, and misplaced citations. For example in the first paragraph of the introduction itself:
- Midjourny -> Midjourney
- The citation, Somepalli et al. seems misplaced. The paper has nothing to do with the sentence.

Overall, the paper would require a significant rewrite.

2. It's unclear how to understand the proposed metrics for ConceptBench used by the paper.
	- What would qualify as a good M-score or CLIP score after forgetting? Conceptually while decrease in the score is good, it's unclear if too much decrease implies catastrophic forgetting with respect to the concept.
	- How many samples are generated from the diffusion models to compute these scores?
	- How is it measured that the model doesn't forget concepts "drastically"? For example forgetting "Elon Musk" should still generate a human with face unlike Elon Musk. As far as I can understand, none of the metrics capture coherent generations after forgetting a concept.

**Questions:**

1. Are there fundamental issues that can arise by using attention resteering loss? Does the generation's alignment to any prompt start degrading when multiple concepts are forgotten?
2. How is model performance such as CLIP accuracy on similar concepts after removing a concept? This metric can be used to ensure the model fidelity is not deteriorated [1].

[1] Kumari, Nupur, et al. "Ablating concepts in text-to-image diffusion models." _Proceedings of the IEEE/CVF International Conference on Computer Vision_. 2023.